# Highly Efficient Photocatalytic Cr(VI) Reduction by Lead Molybdate Wrapped with D-A Conjugated Polymer under Visible Light

**Ding Liu [†], Yin Wang [†], Xiao Xu, Yonggang Xiang, Zixin Yang * and Pei Wang ***

College of Science, Huazhong Agricultural University, Wuhan 430070, China; liu2649673@sina.com (D.L.); yinwang92@hotmail.com (Y.W.); xuxiao2016hzau@163.com (X.X.); ygxiang@mail.hzau.edu.cn (Y.X.)
* Correspondence: zixinyang@mail.hzau.edu.cn (Z.Y.); peiwang@mail.hzau.edu.cn (P.W.)
† These authors contributed equally to this work.

**Abstract:** Well-designed composite photocatalysts are of increasing concern due to their enhanced catalytic performance compared to a single component. Here, a photocatalyst composed of $PbMoO_4$ (PMO) and poly-benzothiadiazole (BBT, a D-A-conjugated polymer) was successfully synthesized by BBT polymerization on the surface of the PMO. The resultant BBT-PMO with a heterojunction structure represented an enhanced ability to reduce highly toxic heavy metal Cr(VI) from water under visible light irradiation. The 16.7% BBT-PMO(N, nanoscale) showed the best performance. The corresponding $k_{obs}$ over the 16.7% BBT-PMO(N) was 26-fold (or 53-fold) of that over the pure BBT (or pristine PMO(N)), and this activity was maintained after four cycles. The reasons for its good performance are discussed in detail based on the experimental results. Moreover, the synthesis of the BBT in situ of the PMO also altered the morphology of the BBT component, increasing the specific surface area of the BBT-PMO(N) and endowing it with the ability to adsorb Cr(VI). Additionally, the photocatalyst was also environmentally friendly as such a wrapped structure could sustain the high stability of the PMO without dissociation. This work provides a good strategy for efficient photocatalytic Cr(VI) reduction by designing an organic–inorganic hybrid system with high redox capacity.

**Keywords:** photocatalysis; Cr(VI) reduction; D-A-conjugated polymer; $PbMoO_4$; heterojunction

## 1. Introduction

With the rapid development of industrialization, urbanization, and growth of the world's population, environmental pollution attributed to the excessive exploitation and unconscionable use of resources, with unbridled emission of pollutants putting human health at risk for diseases, has attracted significant concern globally [1,2]. Among various kinds of pollutions, the hexavalent heavy metal of hexavalent chromium ion (Cr(VI)) is one of the most common environmental contaminants both in water and air from electroplating, dyeing, and tanning industries, which is non-biodegradable and easily absorbed by the human body, increasing carcinogenic risk [3]. Consequently, many efforts have gone into reducing the harmful effects of long-term residual Cr(VI) in the environment, which is performed by electrolytic, chemical, or photocatalytic reduction of toxic Cr(VI) to a metabolizable Cr(III) [4–6]. Among these technologies, photocatalysis is very eco-friendly to produce photogenerated carriers (electrons and holes) that in turn catalyze oxidation or reduction reactions [5]. However, there are still challenges in photocatalysis, such as fast charge recombination and limited visible light absorption, in practical application for pollutant reduction [7–12].

The construction of heterojunction photocatalysts provides an opportunity to overcome the aforesaid challenges [5,8–10,13–22]. Well-designed heterostructured photocatalysts can have a broader solar spectral response and photogenerated more carriers for

redox reactions than any single semiconductor [9,10,13–22]. Among heterojunction photo-catalysts, those composed of organic semiconductors and inorganic semiconductors are particularly appealing due to their convenient preparation and efficient photo-carrier separation [13,15,23–25]. Donor-acceptor (D-A)-conjugated polymers are an emerging class of organic semiconductors, which consist of alternating electron-rich and electron-poor conjugated units with delocalized $\pi$-conjugation and thus exhibited photocatalytic redox ability. Due to their adjustable molecular backbones, D-A-conjugated polymers have tunable Highest Occupied Molecular Orbital (HOMO) and Lowest Unoccupied Molecular Orbital (LUMO) positions and thus tend to have a satisfactory light absorption capacity [7,13,23,24]. Poly-benzothiadiazole (BBT) is one kind of D-A-conjugated polymer and easily produces photogenerated carriers under visible light for its narrow bandgap (ca. 2.1 V) [7,13,23,25]. However, with fast carrier recombination, BBT merely exhibits a strong singlet oxygen generation capacity and its carriers cannot participate directly in the redox reaction with the substrate. Based on the previously reported literature, the photocatalytic activities of BBT could be much improved by introducing some inorganic semiconductors, i.e., $TiO_2$ and $Bi_2MoO_6$ [13,25]. For example, a Z-scheme heterojunction of BBT and $Bi_2MoO_6$ is reported with enhanced activity in photocatalytic Cr(VI) reduction under visible light, which was a 5-fold rate constant of BBT [25]. In this Z-scheme heterojunction, the fast carrier recombination of BBT was suppressed by neutralizing the holes in the HOMO of BBT with the electrons from the other component, $Bi_2MoO_6$ [25]. Correspondingly, if the inhibition of fast carrier recombination is achieved by transferring the electrons away from the LUMO of BBT to the other component of the heterojunction, the effect on the photocatalytic reduction of Cr(VI) over the heterojunction remains obscured.

Lead molybdate ($PbMoO_4$, PMO) occurs in nature as an abundant mineral called "Wulfenite". It has a wide bandgap and relatively strong redox ability under ultraviolet light and was developed as a low-cost photocatalyst material. However, PMO has little photocatalytic activity under visible light, thus hampering the practical application of this material [26,27]. To extend its light responding range, PMO is often combined with visible light-absorbing materials. Therefore, constructing a heterojunction of BBT and PMO could be a way to endow the heterojunction with an enhanced photocatalytic ability for Cr(VI) reduction under visible light.

To validate the above strategy, nanoscale and micron-scale lead molybdate (PMO(N) and PMO(M)) were prepared as shown in Scheme 1, and both were wrapped with BBT via the in situ synthetic cladding method. Combining multiple characterizations, PMO wrapped with BBT, i.e., the BBT-PMO heterojunction, showed physicochemical advantages compared to the components for Cr(VI) photoreduction under visible light. Especially, the 16.7% BBT-PMO(N) possessed a reaction rate constant 26-fold of that over BBT, which was much better than heterojunctions reported based on BBT [15,25]. It also exhibited superior stability, with more than 90% photoreduction activity after four cycles. Under visible light irradiation, the outwear of BBT could respond to irradiation and generate photoelectrons and then transfer them to the inner PMO, which promotes the separation of photogenerated carriers. Such a structure of BBT over PMO was also found to prevent Pb from releasing through photocorrosion. This strategy provides a possible pathway to construct an organic-inorganic hybrid system with high redox capacity for efficient photocatalytic Cr(VI) reduction.

## 2. Results and Discussion

### 2.1. Photoreduction and Adsorption Activity of Cr(VI) over Materials

Photocatalytic reduction of Cr(VI) over pure BBT, pristine PMO, BBT-PMO, and BBT+PMO (a fine physically mechanical mixture of pure BBT and pristine PMO) was performed under visible light. Figure 1A and B depict the normalized change of Cr(VI) concentrations ($c_t/c_0$) over time during the photocatalytic process, which began after an absorption equilibrium in the dark. Pure BBT, pristine PMO(M&N), and their physically mechanical mixtures (13.3% BBT+PMO(M) and 16.7% BBT+PMO(N)) all showed a poor

ability to reduce Cr(VI). The photocatalytic activity of the BBT-PMO(M) increased gradually with the BBT nominal mass ratio rising to 13.3%, and then decreased gradually with the above ratio up to 20%. Similar trends were found for the BBT-PMO(N), while 16.7% BBT-PMO(N) showed the best performance.

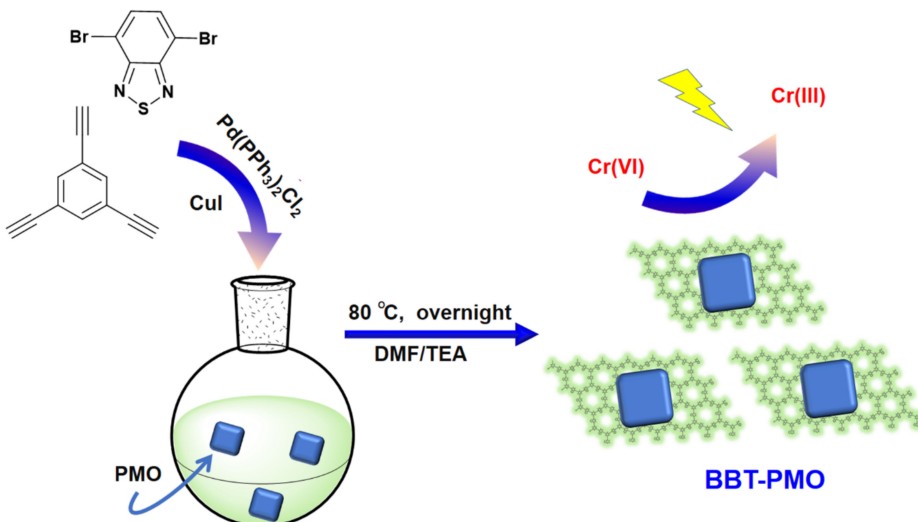

**Scheme 1.** The synthesis of the BBT-PMO heterojunction and the photocatalytic Cr(VI) reduction over the heterojunction.

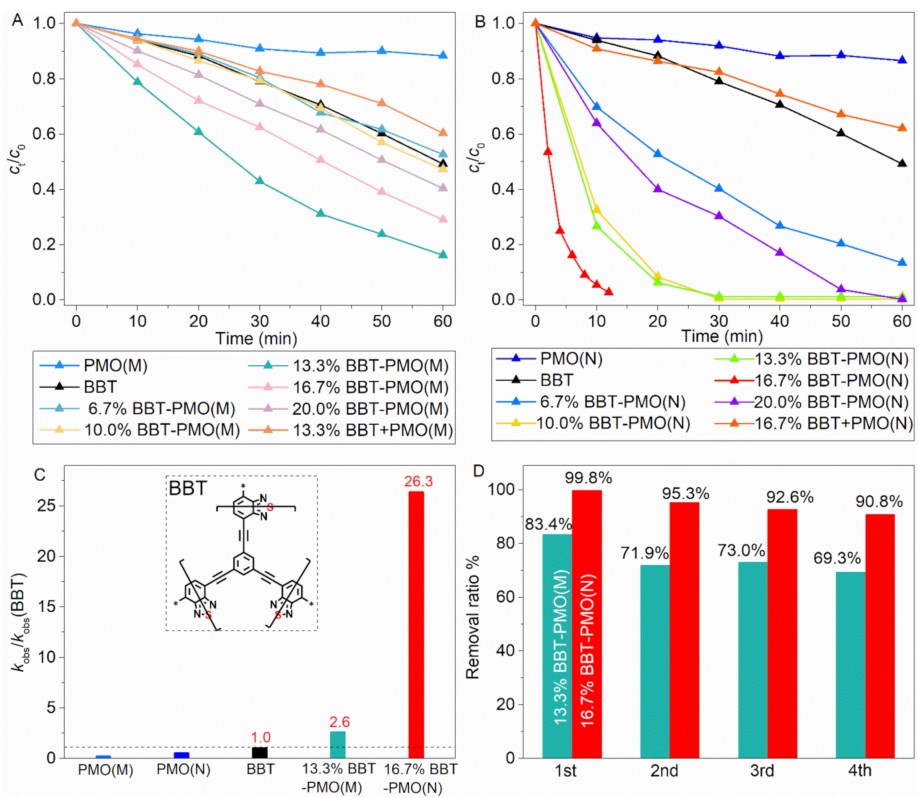

**Figure 1.** Cr(VI) photocatalytic reduction under visible light over (**A**) x% BBT-PMO(M), BBT, PMO(M), and 13.3% BBT+PMO(M), and (**B**) x% BBT-PMO(N), BBT, PMO(N), and 16.7% BBT-PMO(N) (x = 6.7, 10.0, 13.3, 16.7, 20.0). (**C**) The ratio of reaction rate constants for PMO(M), PMO(N), BBT, 13.3% BBT-PMO(M), and 16.7% BBT-PMO(N) compared to pure BBT, with insertion of the BBT monomer structure. (**D**) The cycling runs for the photoreduction of Cr(VI) on 13.3% BBT-PMO(M) (cyan) and 16.7% BBT-PMO(N) (red).

The kinetic curves of the photocatalytic activity for all samples are fitted with a pseudo-first-order kinetic equation:

$$ln\ (c_t/c_0)\ =\ -k_{obs} \times t \tag{1}$$

where $k_{obs}$ is the reaction rate constant, $c_t/c_0$ is the residual Cr(VI) relative concentration, and $t$ is the irradiation time [28,29]. Table 1 displays the value of $k_{obs}$ for all samples. It was found that 16.7% BBT-PMO(N) (0.316 $min^{-1}$) or 13.3% BBT-PMO(M) (0.031 $min^{-1}$) showed the highest Cr(VI) to Cr(III) rate constant in BBT-PMO(N) series or in BBT-PMO(M) series, respectively. The reaction rate constant of 16.7% BBT-PMO(N) is 10 times greater than that of 13.3 % BBT-PMO(M) (Figure 1C). Significantly, the reaction rate constant of 16.7% BBT-PMO(N) possesses a 26-times greater value than that of the pure BBT (0.012 $min^{-1}$) (Figure 1C). Furthermore, as a control experiment, PMO(N) was etched away from 16.7% BBT-PMO(N) by 2 M $HNO_3$, the adsorption ability to Cr(VI) of the rest material after etching remained unchanged, but the photocatalytic reduction performance was greatly reduced to the low level of the BBT. Therefore, the formation of the heterojunction is very critical to improve photocatalytic performance. The activity for Cr(VI) photoreduction over the PMO and the BBT was significantly enhanced once they were combined into a heterojunction, under visible light irradiation. In addition, the particle size of PMO in the heterojunction was also a key factor to be considered.

**Table 1.** The reaction rate constants ($k_{obs}$ ($min^{-1}$)) for Cr(VI) photocatalytic reduction on different photocatalysts, which was determined by the first-order kinetic model.

| | $s$ = M | $s$ = N |
|---|---|---|
| PMO($s$) | 0.002 | 0.006 |
| BBT | 0.012 | 0.012 |
| | | 0.010 * |
| 6.7% BBT-PMO($s$) | 0.011 | 0.03 |
| 10.0% BBT-PMO($s$) | 0.012 | 0.183 |
| 13.3% BBT-PMO($s$) | 0.031 | 0.15 |
| 16.7% BBT-PMO($s$) | 0.02 | 0.316 |
| 20.0% BBT-PMO($s$) | 0.015 | 0.041 |
| $x$% BBT+PMO($s$) | 0.008 ($x$ = 13.3) | 0.006 ($x$ = 16.7) |

* 16.7% BBT-PMO(N) after etching the PMO(N) component by 2 M $HNO_3$.

In terms of cyclic stability, 13.3% BBT-PMO(M) and 16.7% BBT-PMO(N) were examined for Cr(VI) removal ability in four photocatalytic cycles, as shown in Figure 1D. After four cycles, the Cr (VI) removal ratio for 13.3% BBT-PMO(M) decreased to about 70%, and the value remained about 90% for 16.7% BBT-PMO(N). This indicated the superior stability of 16.7% BBT-PMO(N). The regenerated 16.7% BBT-PMO(N) and 13.3% BBT-PMO(M) continued to have a significantly higher performance than that of the PMO and the BBT.

Photocatalytic reactions often occur on the catalyst surface, and thus the ability of the photocatalyst to adsorb the substrate also affects its catalytic performance. During the dark adsorption step before the photocatalytic process, the BBT and the PMO exhibited little capacity to adsorb Cr(VI), while the BBT-PMO with high BBT content presented a good capacity. Thus, the Cr(VI) adsorption experiment was also performed to investigate the maximum adsorption capacities of 16.7% BBT-PMO(N) and 13.3% BBT-PMO(M). The Langmuir model (Figure 2) was carried out to analyze the adsorption data, with the relevant parameters summarized in Figure 2.

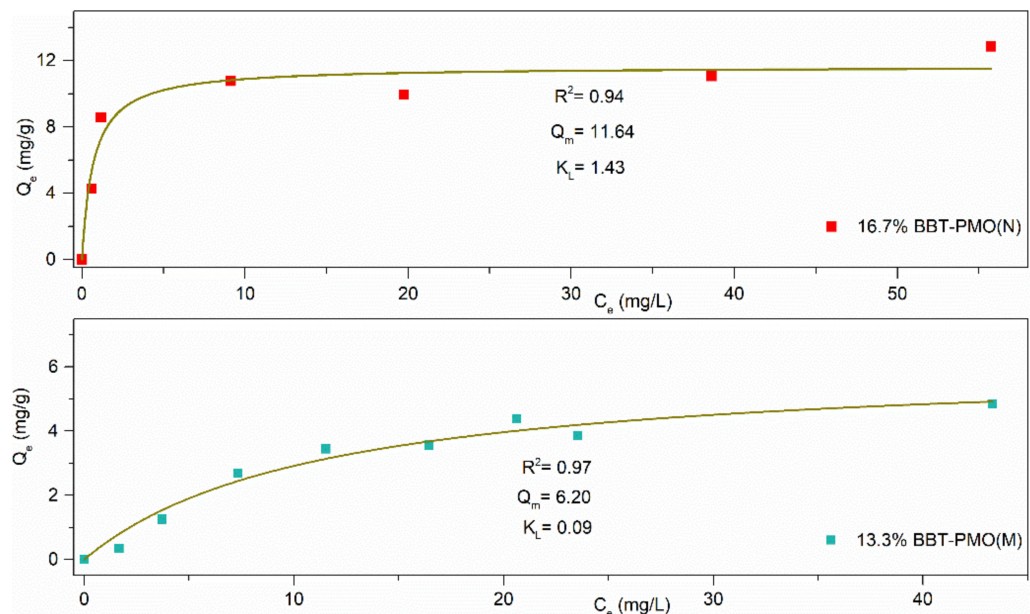

**Figure 2.** Adsorption kinetics of Cr(VI) for 16.7% BBT-PMO(N) (top) and 13.3% BBT-PMO(M) (bottom) at 298 K, pH = 2, inserted with the fitting kinetic parameters.

The Langmuir adsorption isotherm equation is:

$$C_e/Q_e = 1/(Q_m K_L) + C_e/Q_m \qquad (2)$$

where $Q_e$ (mg/g) is referred to as the amount of equilibrium adsorption, $C_e$ (mg/L) is the equilibrium concentration, $Q_m$ represents the maximum adsorption capacity, and $K_L$ is the Langmuir constant [30].

As shown in Figure 2, the adsorption amount increased when the Cr(VI) concentration increased, and the adsorption of Cr(VI) on 16.7% BBT-PMO(N) and 13.3% BBT-PMO(M) was fit with a one-layer adsorption model. Based on the Langmuir adsorption isotherm equation, the obtained maximum adsorption capacity at 298 K was 11.64 mg/g for 16.7% BBT-PMO(N) and was about 1.9 times higher than 13.3% BBT-PMO(M) with that value being 6.20 mg/L. This indicates that the loading of Cr(VI) was higher on 16.7% BBT-PMO(N) than that of 13.3% BBT-PMO(M).

## 2.2. Structure and Composition of Materials

Based on the above photoreduction activity analysis, 13.3% BBT-PMO(M) and 16.7% BBT-PMO(N) had an outstanding performance for Cr(VI) reduction. Therefore, we chose these two materials as representatives to carry out the structure and composition study of the BBT-PMO heterojunction. The purity and crystallinity of as-synthesized PMO, BBT, and BBT-PMO were well confirmed by XRD analysis. As shown in Figure 3, the achieved XRD peaks were indexed to the tetragonal crystal PMO system, corresponding to Joint Committee on Powder Diffraction Standards (JCPDS) 74-1075. The peaks of the PMO still exist after the process of coating, indicating no impact on the particle size and the crystallization of PMO components. There were no impurity-related peaks and these exhibited perfectly similar plane indexing after BBT loading, for 13.3% BBT-PMO(M) and 16.7% BBT-PMO(N). This indicated that the particle size and crystallization of PMO components had barely been affected by BBT loading. The curve of BBT displayed a broad peak, which was attributed to an amorphous state. Furthermore, this also leads to barely diffracted peaks of the BBT composites for 13.3% BBT-PMO(M) and 16.7% BBT-PMO(N).

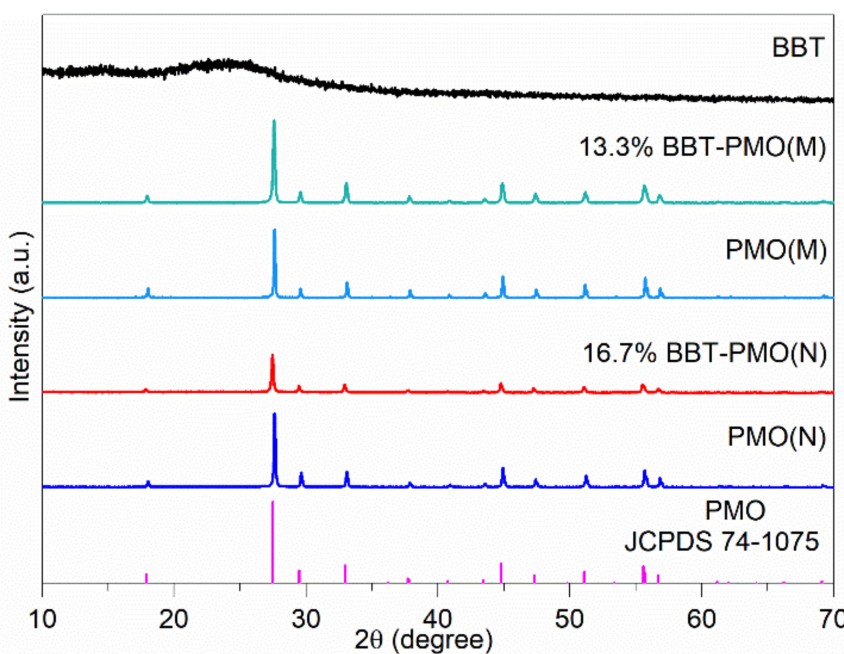

**Figure 3.** XRD patterns of BBT, 13.3% BBT-PMO(M) and PMO(M), and 16.7% BBT-PMO(N) and PMO(N).

The existence of the BBT component in the BBT-PMO was confirmed by Raman spectroscopy. As shown in Figure 4, the characteristic peaks of BBT at 1364 and 1537 cm$^{-1}$ corresponded to the C–C skeleton stretching vibration and that of the aromatic ring in BBT, respectively [7,13]. The peaks around 260, 756, and 860 cm$^{-1}$ could be assigned to the bending vibrations of different components of the MoO$_4$ tetrahedron [25]. The peaks of the BBT-PMO sample covered both BBT and PMO. This indicated that BBT was successfully loaded onto the PMO powder, which was verified to be a heterojunction form in the following analysis.

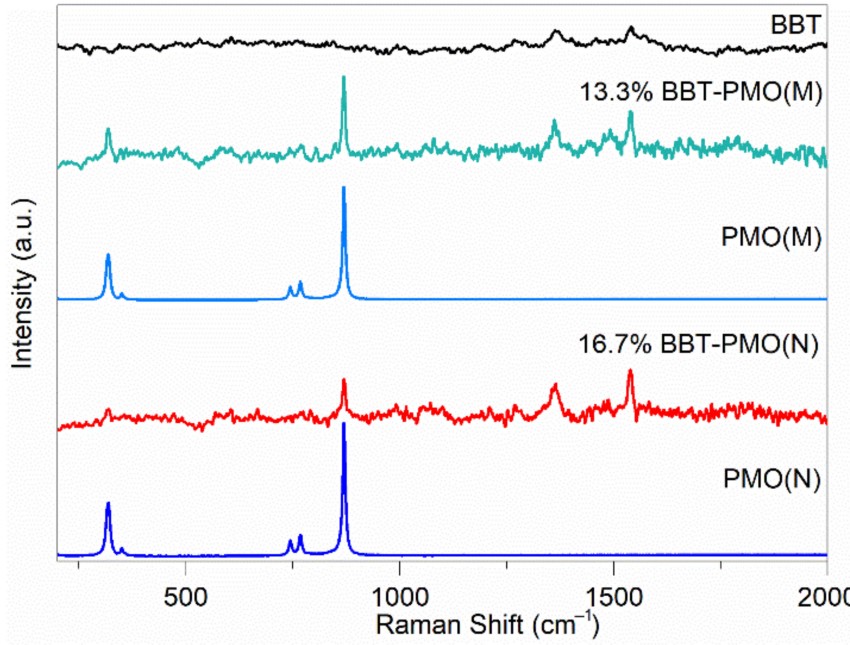

**Figure 4.** Raman spectra of BBT, 13.3% BBT-PMO(M) and PMO(M), and 16.7% BBT-PMO(N), and PMO(N).

The microstructure and morphology of these samples were characterized by TEM and HRTEM. Figure 5A,C demonstrated that the PMO(M) presented a blocky structure with a uniform size of 5–10 μm, while the PMO(N) exhibited similar morphology to PMO(M) with a much smaller size of 100 nm. Figure 5B,D displayed TEM images of BBT-PMO with the in situ polymerization of BBT. PMO(M) was surrounded by irregular flocculent BBT for 13.3% BBT-PMO(M), while PMO(N) presented a state evenly wrapped by BBT for 16.7% BBT-PMO(N). Furthermore, 13.3% BBT-PMO(M) and 16.7% BBT-PMO(N) were also analyzed by HRTEM to further confirm the detailed structure. The clear lattice distances between two adjacent interplanar, as shown in Figure 5E,F, were both 0.324 nm, and assigned to the (112) facet of PMO for both 13.3% BBT-PMO(M) and 16.7% BBT-PMO(N). The shallow area without a lattice fringe could be attributed to the amorphous BBT for 13.3% BBT-PMO(M), while 16.7% BBT-PMO(N) showed a well-proportioned coating of PMO(N) wrapped by BBT. This suggested that the particle size of the PMO had a key influence on the morphology of the photocatalysts.

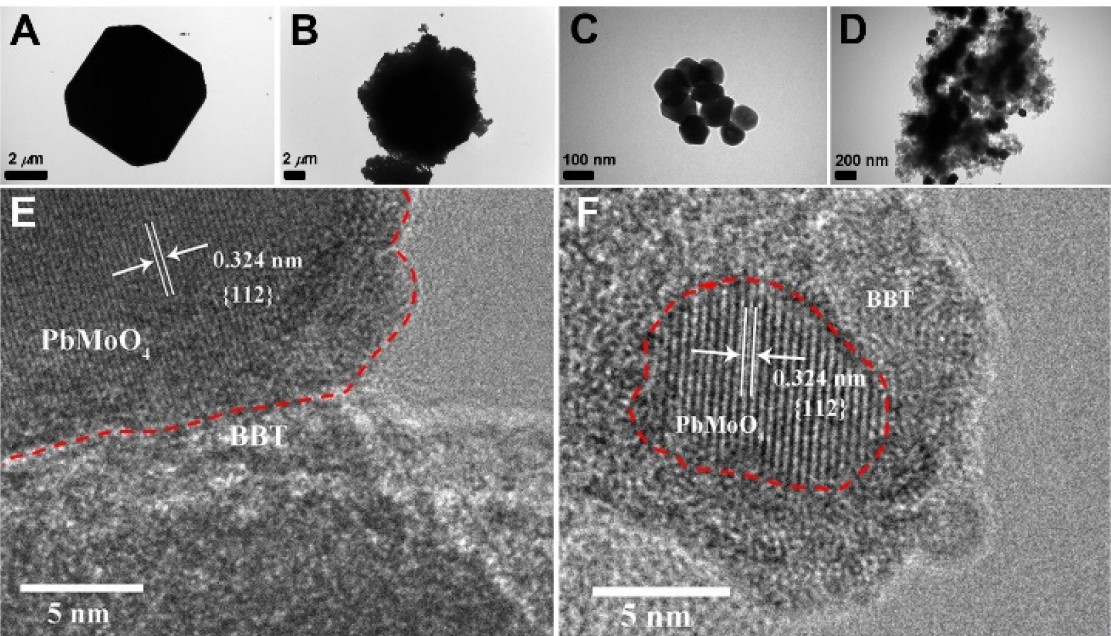

**Figure 5.** TEM images of (**A**) PMO(M), (**B**) 13.3% BBT-PMO(M), (**C**) PMO(N), and (**D**) 16.7% BBT-PMO(N). HRTEM images of (**E**) 13.3% BBT-PMO(M), and (**F**) 16.7% BBT-PMO(N).

Interestingly, the introduction of the PMO particles during the polymerization of the BBT significantly increases its surface area. By BET measurement [7,15], the specific surface areas were 8.65 and 77.76 m$^2$/g of PMO(N) and 16.7% BBT-PMO(N), respectively. The 16.7% BBT-PMO(N) exhibited an about 4 times larger surface area than pure BBT, as shown in Table 2. This was attributed to the well-dispersed BBT wrapped on PMO(N). Although the specific surface area of 13.3% BBT-PMO(M) was smaller than that of the pure BBT, the dispersion of the BBT on the PMO(M) was improved after in situ synthesis. In fact, the BBT component was only a small amount in the heterojunction and the specific surface of the pristine PMO was extremely small. The deduction for BBT well-dispersed on PMO was consistent with the TEM results, where BBT showed loose morphology in BBT-PMO. Thus, the BBT component in the heterojunction encapsulated the PMO component inside and had a loose and large surface, which facilitated the BBT component to more easily bind reaction substrates such as Cr(VI) and more efficiently sensitize and protect the PMO component.

**Table 2.** Specific surface area (S, unit: $m^2/g$) of 13.3% BBT-PMO(M), PMO (M), BBT, 16.7% BBT-PMO(N), PMO(N), and of BBT-PMO(N).

|  | BBT | PMO(M) | 13.3% BBT-PMO(M) | PMO(N) | 16.7% BBT-PMO(N) |
|---|---|---|---|---|---|
| $S$ | 18.69 | 2.29 | 6.17 | 8.65 | 77.76 |

Thermogravimetric analysis (TGA) was conducted for the PMO, 13.3% BBT-PMO(M), and 16.7% BBT-PMO(N) as shown in Figure 6. For the PMO(M) or the PMO(N), a weight loss lower than 0.5 wt% occurred from 100 °C, which could be attributed to the loss of $H_2O$, and it keeps stable when temperature continued rising. For 13.3% BBT-PMO(M) and 16.7% BBT-PMO(N), a weight loss occurred from 300 to 500 °C, which could be attributed to the thermo-decomposition of thiadiazol from the BBT [7,13]. Notably, the real mass ratio of the BBT component in the BBT-PMO could be calculated based on the TGA results. The real loading amount of BBT was 10.2 and 14.2 wt% for 13.3% BBT-PMO(M) and 16.7% BBT-PMO(N), respectively, which was approximate to its nominal mass ratio.

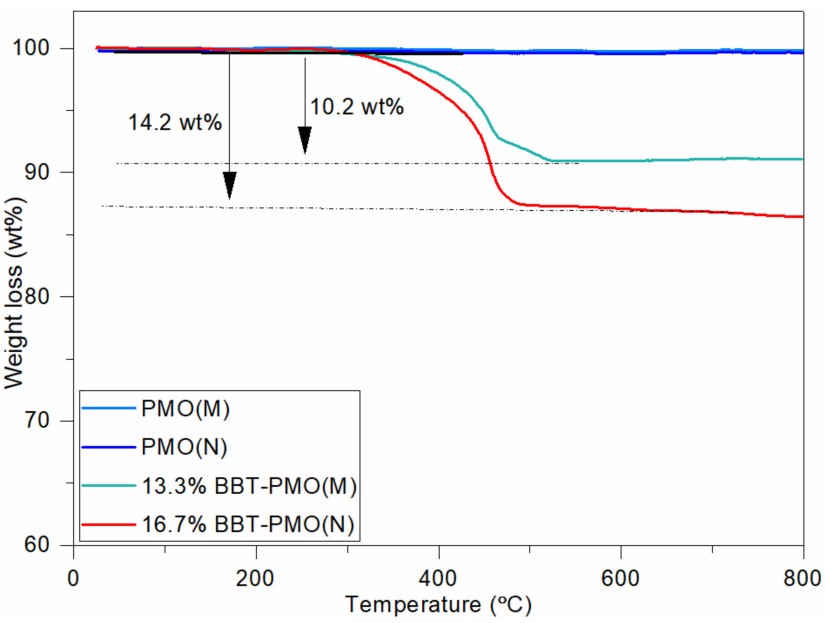

**Figure 6.** TGA of PMO(M), PMO(N), 13.3% BBT-PMO(M), and 16.7% BBT-PMO(N).

Furthermore, the amount of dissolved Pb which was released from materials during the photocatalytic reaction was also detected by ICP-MS, since Pb was also a common toxic heavy metal. It was found that the concentration of dissolved Pb in the reaction solution was 0.069 or 0.099 mg/L after four photocatalytic cycles for 16.7% BBT-PMO(N) or for 13.3% BBT-PMO(M), respectively, which are in the safe Pb concentration range ($\leq$0.1 mg/L) for aquatic life, barely revealing the emergence of Pb pollutants from the heterojunction [31,32]. However, the concentration of dissolved Pb for the PMO during the photocatalytic reaction was 0.721 mg/L. These results indicate that the BBT component is helpful to alleviate the photo corrosion of the PMO component in the heterojunction.

### 2.3. Mechanism of Photoreduction of Cr(VI) over Material

The recombination processes of electron-hole pairs in the photocatalytic system have important impacts on its photocatalytic ability. Photocurrent analysis was applied to investigate the photogenerated charge separation capacity and the electron transfer dynamics among BBT, PMO, 13.3% BBT-PMO(M), and 16.7% BBT-PMO(N). As shown in Figure 7, under visible light irradiation, the five samples exhibited a quickly enhanced photocurrent response and remain constant at a comparable high value. In contrast, the photocurrent

response intensity remained at a very low value in the dark condition. Moreover, the BBT-PMO composite exhibited higher photocurrent response density compared with the pure BBT and pristine PMO. Furthermore, the 16.7% BBT-PMO(N) presented the highest photocurrent, which was 2.4 times higher than the 13.3% BBT-PMO(M). The giant photocurrent suggested the 16.7% BBT-PMO(N) obtained greater photogenerated charge separation efficiency, which could lead to higher photocatalytic activity as well. The enhanced photocurrent in the 16.7% BBT-PMO(N) and the 13.3% BBT-PMO(M) were comparable to the BBT and the PMO, indicating that the heterojunction forms decreased the recombination of photogenerated electron-hole pairs, and therefore more electrons could participate in the photoreduction reaction.

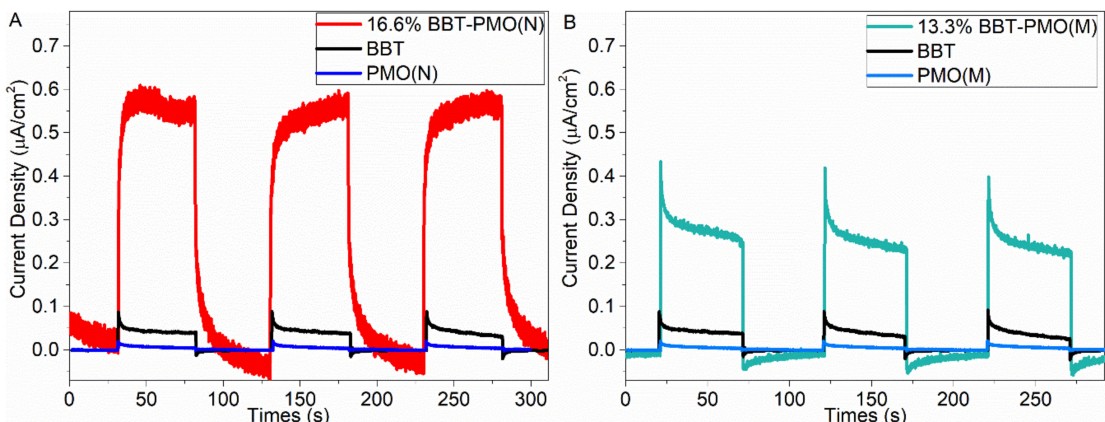

**Figure 7.** Photocurrent measurement of (**A**) BBT, PMO(N) and 16.7% BBT-PMO(N), and (**B**) BBT, PMO(M) and 13.3% BBT-PMO(M).

To probe the reason for the enhanced photoactivity, the surface elemental composition of BBT-PMO and corresponding PMO substrates were further investigated via XPS spectra. High-resolution XPS spectra of Mo 3d, Pb 4f, and O 1s are displayed in Figure 8. The binding energy of Mo 3d and Pb 4f in PMO did not change after being wrapped by BBT. However, compared to the spectrum of the pristine PMO, a new peak was detected for 13.3% BBT-PMO(M) and 16.7% BBT-PMO(N) at 533.12 or at 533.56 eV, which was attributed to the O 1s in the O–C bond [25]. This indicated that the carbon atoms in the BBT might interact with the oxygen atoms in the PMO at the interface of the two components. This new O–C bond formed after BBT loading on PMO was a sign of the BBT-PMO heterojunction formation. The electronic bonding energy of the O–Pb bond changed from 532.4 to 532.0 eV after 16.7% BBT was loaded onto the PMO(N), while it was barely changed for 13.3% BBT-PMO(M) compared to PMO(M) [27]. Likewise, the peak of the O–Mo bond was also changed from 530.05 to 530.2 eV after 16.7% BBT was loaded onto the PMO(N), while it was not changed for 13.3% BBT-PMO(M) compared to PMO(M) [27]. These experimental results showed that the BBT component had a more significant effect on the chemical valence state of the surface elements of the PMO(N) than that of the PMO(M) after the formation of the heterojunction, i.e., the BBT and the PMO(N) could form a tighter heterojunction, which might be related to the smaller size and the larger specific surface area of the PMO(N) than the PMO(M).

To further elucidate that the electron-hole recombination rate is lower in the BBT-PMO than in the BBT, the position of the energy band in the materials was examined. DRS spectra were assessed to explore the optical properties and bandgaps. Figure 9 exhibits the representative absorption spectrum of the BBT, the PMO, the 13.3% BBT-PMO(M), and the 16.7% BBT-PMO(N). It was demonstrated that the adsorption band edges of PMO(M) or PMO(N) were located around 400 nm. This indicated a strong absorption in the UV light region, not the visible light region, for the pristine PMO. In contrast, BBT obtained an adsorption band edge around 600 nm, indicating a strong light absorption capacity

in both the UV and visible regions. After loading of the BBT upon the PMO, the light absorption range of the BBT-PMO displayed a wide visible light absorption compared to the pristine PMO. The bandgap of the semiconductor was calculated from the corresponding adsorption band edge by using the Kubelka-Munk formula, as shown in Figure 9. The bandgaps ($E_g$) for BBT, PMO(M), and PMO(N) were determined to be 2.11, 3.08, and 3.16 eV, respectively [7,33]. The bandgap calculated above was consistent with the HOMO and the LUMO position of the BBT, which has been reported as +1.43 and −0.68 eV (vs. NHE) [7]. As for PMO, it could not be activated under light irradiation. Therefore, the PMO(M) and the PMO(N) showed almost no photocurrent under visible light irradiation. The light adsorption capacity of the heterojunction was enhanced with BBT loading, and thus might increase the Cr(VI) photoreduction over 13.3% BBT-PMO(M) and 16.7% BBT-PMO(N) under visible light.

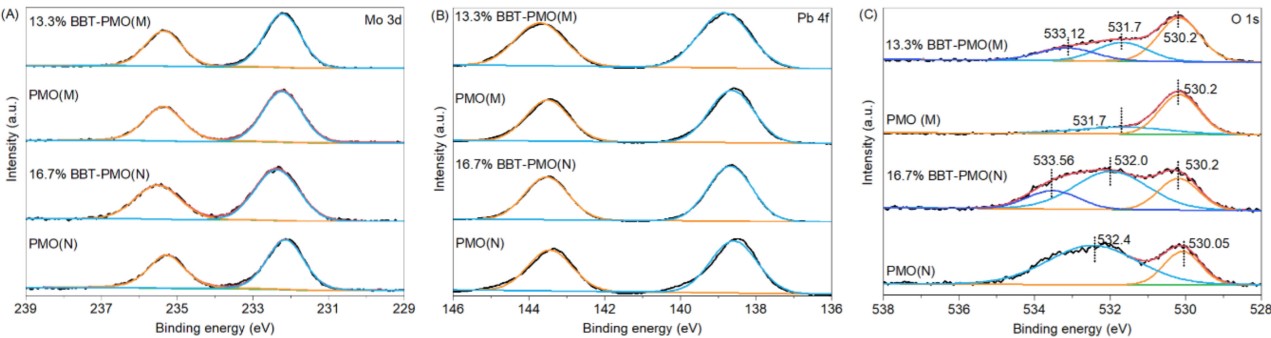

**Figure 8.** XPS spectra of (**A**) Mo 3d, (**B**) Pb 4f, and (**C**) O 1s in PMO(M), 13.3% BBT-PMO(M) and PMO(N) and 16.7% BBT-PMO(N).

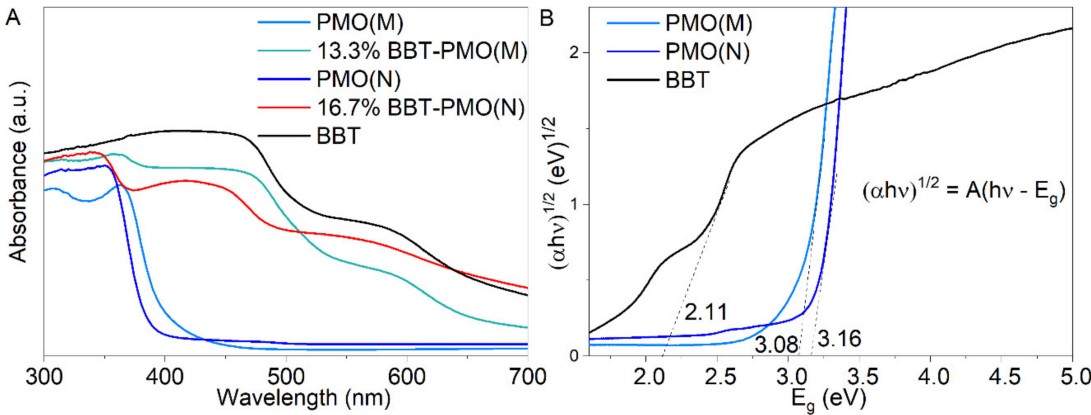

**Figure 9.** (**A**) UV-vis DRS of BBT, PMO(M), 13.3% BBT-PMO(M), PMO(N), and 16.7% BBT-PMO(N). (**B**) Tauc plots of BBT, PMO(M), and PMO(N).

The positions of the conduction band (CB) and the valence band (VB) of PMO were calculated from the empirical formulas below:

$$E_{VB} = X - 4.50 + 0.5E_g \tag{3}$$

$$E_{CB} = E_{VB} - E_g \tag{4}$$

where $X$ is the Mulliken electronegativity of PMO with a value of 5.98 eV, and $E_{VB}$ and $E_{CB}$ are both vs. NHE [33]. Thus, the $E_{CB}$ and $E_{VB}$ were calculated to be +3.02 and −0.08 eV for the PMO(M); and +3.06 and −0.10 eV for the PMO(N). Possessing a LUMO with a relatively negative potential of −0.68 eV, the BBT was an electronic donor in the photocatalytic process [7,15,25]; that is, the BBT could transfer electrons to the CB of the

PMO (ca. $-0.10$ eV) and then donate electrons into the photoreduction reaction. As a result, the photogenerated carrier's separation and transportation of the BBT were promoted, and the photocatalytic activity of the BBT-PMO was increased consequently.

Moreover, the pathway of the photocatalytic reduction of Cr(VI) over the BBT-PMO was investigated. Generally, the electron ($e^-$) and the superoxide anion ($^\bullet O_2^-$) are expected to be the active species for the photoreduction of Cr (VI). To determine the possible contribution of $^\bullet O_2^-$, $N_2$ was bubbled into the solution to exclude the dissolved oxygen to interdict $^\bullet O_2^-$ formation during the photocatalytic reaction. As shown in Figure 10A, the reduction kinetic constants over the 16.7% BBT-PMO(N) and the 13.3% BBT-PMO(M) were decreased by 43.6% and 35.7%, respectively. These results indicated $e^-$ and $^\bullet O_2^-$ were both the important reactive species for the photoreduction to Cr(VI) in the BBT-PMO system. The corresponding EPR analysis (Figure 10B) also confirmed the generation of $^\bullet O_2^-$ in the photocatalytic systems when the 16.7% BBT-PMO(N) and the 13.3% BBT-PMO(M) were excited by visible light [25]. When the PMO was irradiated with UV light, the $^\bullet O_2^-$ was not detected in the photocatalytic system. Because the CB of the PMO was too positive to reduce the oxygen and generate the $^\bullet O_2^-$, the PMO was excited under the UV light. Interestingly, the BBT system did not produce the $^\bullet O_2^-$ under visible light, although the LUMO potential of the BBT could thermodynamically reduce oxygen to generate the $^\bullet O_2^-$. This suggested that the electron-hole complexation was inhibited so that oxygen had the opportunity to react with the electrons on its LUMO to generate $^\bullet O_2^-$.

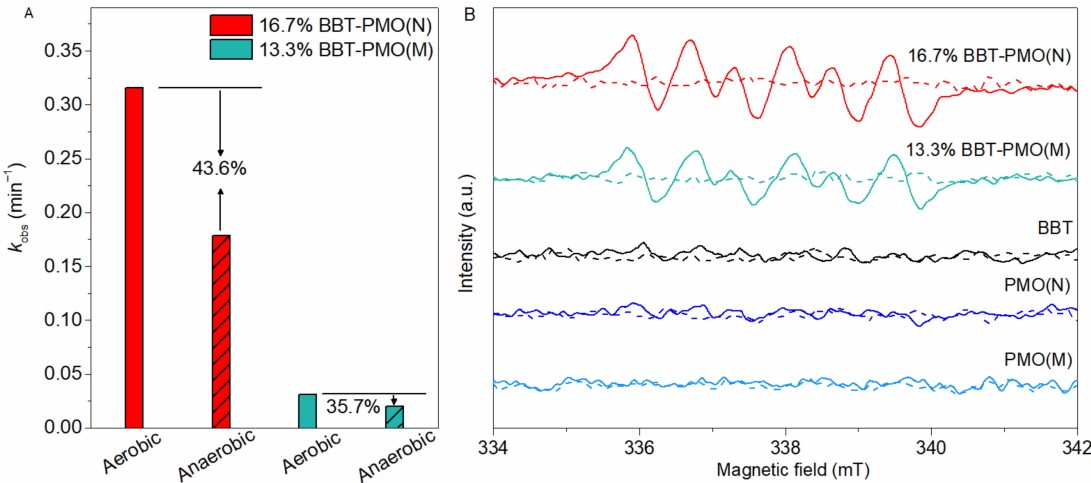

**Figure 10.** (**A**) Photoreduction rate constants of Cr(VI) over 16.7% BBT-PMO(N) and 13.3% BBT-PMO(M) under different atmospheres. (**B**) EPR spin-trapping signals of BBT, PMO(N), 16.7% BBT-PMO(N), PMO(M), and 13.3% BBT-PMO(M) for $^\bullet O_2^-$. The dotted line is for light off and the solid line for the light on.

Based on the above results and analysis, a possible photocatalytic mechanism in the BBT-PMO system under visible light has been proposed as shown in Figure 11 [4,25,34]. The PMO component was not activated under visible light irradiation, while the BBT component was able to generate electrons and holes under visible light. For the heterojunction structure, the photogenerated electrons could easily transfer from the LUMO of BBT to the CB of PMO. Therefore, the recombination of the electrons on the LUMO with the hole on the HOMO was effectively suppressed, and then the electrons had more chance to react with Cr(VI) or oxygen. Furthermore, the hole might be consumed with $H_2O$ to produce $O_2$. That is, the photocatalytic reduction activity of the BBT-PMO system under visible light was effectively enhanced for its heterojunction structure

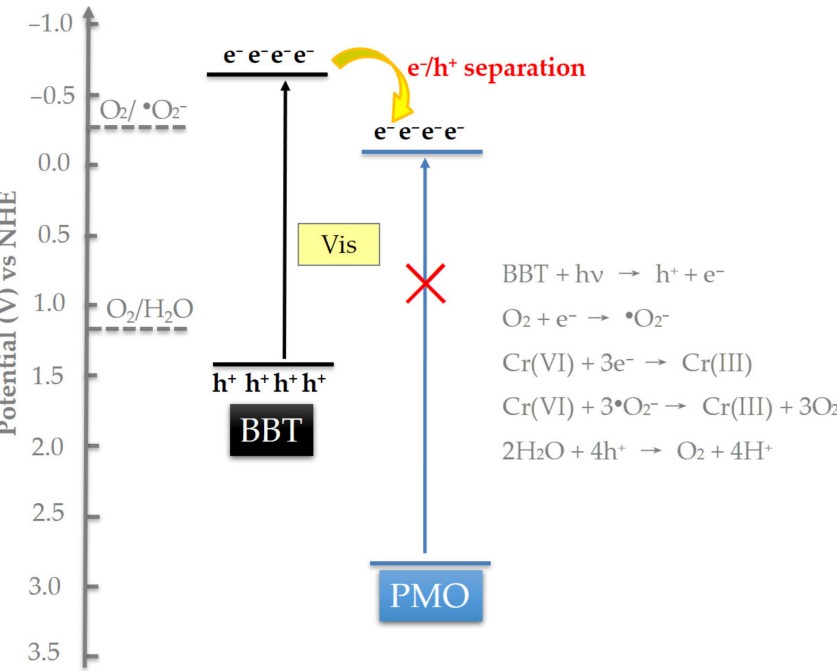

**Figure 11.** The possible reaction mechanism for Cr(VI) reduction over 13.3% BBT-PMO(M) and 16.7% BBT-PMO(N).

## 3. Materials and Methods

### 3.1. Synthesis of Lead Molybdate

All samples were synthesized by a simple solvothermal method. The specific steps are as follows.

For the synthesis of micron lead molybdate (PMO(M)), 0.662 g of lead nitrate was dissolved in 30 mL of 1.0 mol/L nitric acid solution. Then, 0.352 g of ammonium molybdate was dissolved in 10 mL deionized water, which was added to the lead nitrate solution dropwise and further transferred into a 50 mL PTFE lined stainless steel autoclave. After being heated at 160 °C for 8 h, the PMO(M) was collected by centrifugation and washed several times with ethanol and distilled water repeatedly. The resultant powder was calcined at 400 °C for 6 h.

For the synthesis of nanoscale lead molybdate (PMO(N)), 0.18 g of sodium oleate ($C_{18}H_{33}NaO_2$) was dissolved in 60 mL deionized water, and 0.662 g of lead nitrate was added. Then, 0.352 g of ammonium molybdate was dissolved in 10 mL deionized water, which was added to the lead nitrate solution dropwise, further transferred into a 50 mL PTFE lined stainless steel autoclave, and then heated at 160 °C for 8 h. Finally, the PMO(N) was collected by centrifugation, washed several times with ethanol and distilled water repeatedly, and then calcined at 400 °C for 6 h.

### 3.2. Synthesis of BBT-PMO

To prepare the BBT-PMO heterojunction, the BBT was introduced onto the surfaces of suspended PMO(M) or PMO(N) powder via a facile in situ cross-coupling reaction. Firstly, 300 mg of as-prepared PMO(M)/PMO(N) powder was suspended in the mixed solvent DMF/TEA (5 mL/5 mL) in Schlenk tubes; then, 1,3,5-triethynylbenzene and 4,7-dibromobenzo[c][1,2,5]thiadiazols were added into the solution with $Pd(PPh_3)_2Cl_2$ and CuI as catalysts. After vigorously stirring at 80 °C for 24 h, the resulting precipitate was filtered off, washed in a Soxhlet extractor containing a mixed solvent of DCM/MeOH for about 48 h, and dried overnight at 60 °C. Moreover, to investigate the effects of BBT content on the photocatalytic performance of hybrids, a series of samples with different weight ratios were also synthesized and labeled as 6.7%, 10%, 13.3%, 16.7%, and 20% BBT-PMO(M) or BBT-PMO(N). Here, x% represents x wt%, which means the mass ratio of BBT to PMO(M)

or PMO(N) in the case of a complete conversion of precursors during the cross-coupling reaction. Furthermore, the 13.3% BBT sample mechanically mixed with PMO(M), denoted as 13.3% BBT+PMO(M), and the 16.7% BBT sample mechanically mixed with PMO(N) (16.7% BBT+PMO(N)), were fabricated for comparative analysis.

For comparison, the pure conjugated polymer constructed with a phenyl unit and benzothiadiazole, BBT, was prepared through a Sonogashira-Hagihara reaction between the 1,3,5-triethynylbenzene and 4,7-dibromobenzo[c][1,2,5]thiadiazols reported. It is worth noting that the as-prepared BBT was completely insoluble in any general organic solvent investigated.

### 3.3. Characterization

X-ray diffraction (XRD) patterns of the samples were recorded on a Bruker D8 Advance X-ray diffractometer (Bruker, Billerica, MA, USA) with Cu $K_\alpha$ radiation ($\lambda = 1.5418$ nm) and irradiated at a scan rate of 5 °/min in the range of 5°–80°. The X-ray tube voltage and current were set at 45 kV and 50 mA, respectively. The morphology and microstructure of the samples were characterized by scanning electron microscopy (SEM JEOL 6700-F, JEOL Ltd. Tokyo, Japan) and transmission electron microscopy (TEM, JEOL JEM-2010, JEOL Ltd. Tokyo, Japan). The UV–vis diffuse reflectance spectra (DRS) of the prepared samples in the range of 200–800 nm were recorded in absorption mode using a PerkinElmer lambda 650 s UV-vis spectrophotometer (PerkinElmer, Waltham, MA, USA). The photocurrent measurement was performed on an electrochemical system (CHI-660D, Chenhua Ltd., Shanghai, China) in a conventional three-electrode configuration with a Pt foil as the counter electrode and Ag/AgCl (saturated KCl) as the reference electrode. The working electrodes were prepared by the coating method. Then, 10 mg of the as-prepared photocatalyst and 0.5 mL Nafion dispersed in 5 mL absolute ethanol (99.5%) were mixed and sonicated for 30 min. The mixture was then spread on a $2.5 \times 1.0$ cm indium-tin oxide glass substrate and dried in air. A 300 W Xe lamp was used as the light source (($\lambda \geq 420$ nm), and a 0.1 mol/L $Na_2SO_4$ aqueous solution was used as the electrolyte. The surface elemental composition of the sample was analyzed by X-ray photoelectron spectroscopy (XPS), which used an XSAM800 system with a Mg $K_\alpha$ X-ray as the excitation source, and all the binding energies were referenced to the C 1*s* peak at 284.6 eV of the surface amorphous carbon. The Raman spectrum was collected using a confocal microscope, and the wavelength of the excitation laser was 514 nm. The adsorption and desorption isotherms, pore size distribution, and specific surface area of nitrogen were measured at $-196$ °C using a surface area and pore size analyzer. The generation of $\bullet O^{2-}$ was detected by Electron Paramagnetic Resonance (EPR, Magnettech MS-5000, Bruker, Billerica, MA, USA)) with 5,5-dimethyl-1-pyrroline-N-oxide (DMPO, 100 mM in methanol) as the radical spin-trapped reagent under light irradiation.

### 3.4. Photocatalytic Activity Test

The photocatalytic activities of materials were investigated by photoreduction of toxic Cr(VI) to non-toxic Cr(III) in an aqueous solution under a 300 W Xe lamp (PLS SXE300, Perfectlight Inc., Beijing, China) with a cut-off filter ($\lambda \geq 420$ nm), and the average light intensity was 600 mW cm$^{-2}$, which is 6 times that of AM 1.5G irradiance. The photocatalytic reactor was equipped with a circulation water jacket to keep the reaction temperature constant at 25 °C. In a typical procedure of photocatalytic reduction of Cr(VI), 50 mg of the as-prepared photocatalyst was dispersed ultrasonically into a 50 mL aqueous solution with 20 ppm Cr (VI) and 0.01 M HCl for 10 min. Before illumination, the suspension was stirred for 30 min in the dark to establish an adsorption-desorption equilibrium, then about 4 mL aliquot dispersion of the sample was taken every 5 min over 25 min, followed by centrifugation at 10,000 rpm for 5 min to remove the solid. The concentration of Cr(VI) was determined colorimetric at 540 nm using the standard diphenylcarbazide (DPC) method with a Shimadzu UV-vis spectrophotometer (Shimadzu, Kyoto, Japan) [35]. In the same way, the symbols $C_0$ and $C_t$ were used to represent the initial concentration and corresponding

concentration, respectively, at different times during irradiation. Further analysis revealed the photocatalytic reduction of Cr(VI) followed the first-order kinetic model.

## 4. Conclusions

A series of in situ BBT-loaded PMO were synthesized with different PMO size and content, and thus applied to photocatalytic Cr(VI) reduction. The photoexcited electrons of BBT can easily transfer to the PMO due to the heterojunction structure, leading to a suppression of charge recombination in the BBT, which significantly increased photocatalytic activity. After comprehensive analysis, it was concluded that the enhanced activity of BBT-PMO was mainly from its heterojunction structure, and the amount of BBT loading was vital to its activity. The 16.7% BBT-PMO(N), where 16.7 wt% BBT was loaded onto PMO(N), exhibited the highest photocatalytic activity, with 99.8% Cr(VI) reduced in 60 min under visible light irradiation and had the highest reaction rate ($k = 0.316$ min$^{-1}$). Additionally, it achieved high stability after four reaction cycles with 90% Cr(VI) photoreduction, and little Pb$^{2+}$ (<0.1 mg/L) detected in the reaction solution. Hence, it can be concluded that the development of BBT/PMO(N) was a promising candidate for Cr(VI) photoreduction. This strategy provides a possible pathway to construct an organic-inorganic hybrid system with high redox capacity for efficient photocatalytic Cr(VI) reduction.

**Author Contributions:** Conceptualization, P.W. and Z.Y.; methodology, Y.X., X.X. and Y.W.; formal analysis, D.L. and Y.W.; investigation, D.L.; data curation, D.L. and Y.W.; writing—original draft preparation, D.L.; writing—review and editing, P.W. and Z.Y.; supervision, P.W. and Z.Y.; project administration, Z.Y.; funding acquisition, P.W. and Z.Y. All authors have read and agreed to the published version of the manuscript.

**Funding:** This work was funded by the National Natural Science Foundation of China (21902055), and the Fundamental Research Funds for the Central Universities (2662018QD041, 2662019YJ011).

**Institutional Review Board Statement:** Not applicable.

**Informed Consent Statement:** Not applicable.

**Conflicts of Interest:** The authors declare no conflict of interest.

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
