# Peer review of "Highly Efficient Photocatalytic Cr(VI) Reduction by Lead Molybdate Wrapped with D-A Conjugated Polymer under Visible Light"

_catalysts, doi:10.3390/catal11010106_

Round 1
Reviewer 1 Report
The work is well written and the results provided are of interest. I woud only recommend to add more information about the experimental techniques (producer, sensitivity, the temperature range of measurements) used in the research.
Author Response
Response: Thank the reviewers for publishing recommendations. Also, we have added some details for the experimental techniques in the “Materials and Methods part” (page 14).
3.4. Photocatalytic Activity Test
The photocatalytic activities of materials were investigated by photoreduction of toxic Cr (VI) to non-toxic Cr (III) in an aqueous solution under a 300 W Xe lamp (PLS SXE300, Beijing Perfectlight Inc., China) with a cut-off filter (λ ≥ 420 nm), and the average light intensity was 600 mW cm−2, which is 6 times of AM 1.5G Irradiance. The photocatalytic reactor was equipped with a circulation water jacket to keep the reaction temperature constant at 25 °C.

Reviewer 2 Report
Summary:
The authors investigate the organic-inorganic hybrid material consisting of the donor-acceptor conjugated poly-benzothiadiazole and PbMoO4 for the photocatalytic reduction of the aqueous pollutant Cr6+ to Cr3+.
After characterizing the catalyst material via PXRD, electron microscopy, TGA, XPS, UV-VIS, and N2 sorption experiments, the team investigated the photocatalytic properties of the material under illumination taking colorimetric measurements of the formed DPC-Cr-complexes.
There, it was found that the 16.7wt% BBT-PMO-nano composite features a Cr6+-to-Cr3+ reaction rate constant 26 times higher than BBT.
In order to deconvolute the effects of size of PbMoO4, the quality of the interfacial contact between the organic and inorganic component as well as possible Pb leaching, a variety of control experiments were performed.
In conclusion, the interfacial charge transfer of the heterostructure is the critical parameter governing the increased reactivity of the composite catalyst.
Comments:
The presented manuscript is a careful investigation into the catalytic reduction of Cr6+ with effort put into control experiments in order to separate the various effects at play.
The experimental work is solid and mostly supports the conclusions and hypothesis presented in this paper.
The composite is adequately characterized.
Yet, I believe that at several points in the manuscript, the presentation of the data needs to be improved a) support the claims made as well as b) to enhance the overall comprehension by the reader.
My suggestions are as follows:
1. Present the material first, then the catalytic reaction, and finally the mechanistic considerations. That way, the data becomes more accessible as the reader will be familiar with the material.
2. Create an overview schematic for figure 1 to give the reader an easily accessible, visual impression of the processes and materials.
3. Mention the oxidation reaction that accompanies the reduction of Cr6+ to Cr3+.
4. The XPS section should be removed or heavily modified. Probing depth in XPS is usually around 4 nm. In the presented material, the inorganic component seems to be mostly covered by a thick layer of BBT, as seen in Fig. 1 d). Therefore, the probing depth is too shallow to merit any real conclusions, contrary to the definitive statements on the bottom of page 8. If the layer is not too thick, then the metal ions should be visible by XPS and the data should be included in the manuscript. At the moment, the presented conclusion is speculative and not supported by data, which is why it should be removed.
5. The photocurrent experiment should be described in the methods section.
6. The photocurrent experiments should be prioritized when discussing the mechanism.
7. Equation 3 and 4 need to be referenced to known literature. Also, electronegativity is defined for elements/ions, not compounds. Please specify what X is under these circumstances.
8. Please insert the Kubelka Munk formula that was used to interpret Fig. 9.
9. Mention and include a reference for the ESR data of O2-.
10. Please include a comparison of the leached Pb levels to current thresholds in water as defined by the occupational safety and health administration (USA) or REACH (EU).
Minor changes:
See attached manuscript.

Author Response
Comments:
The presented manuscript is a careful investigation into the catalytic reduction of Cr6+ with effort put into control experiments in order to separate the various effects at play.
The experimental work is solid and mostly supports the conclusions and hypothesis presented in this paper.
The composite is adequately characterized.
Yet, I believe that at several points in the manuscript, the presentation of the data needs to be improved a) support the claims made as well as b) to enhance the overall comprehension by the reader.
Response: We thank the reviewer for the constructive suggestion. According to these comments, we have carefully revised our manuscript, and the revisions were highlighted in the text. We hope the following point-by-point responses would solve your concern.
My suggestions are as follows:
1. Present the material first, then the catalytic reaction, and finally the mechanistic considerations. That way, the data becomes more accessible as the reader will be familiar with the material.
Response: Thank the reviewer’s suggestion. The discussion of the ICP-MS experiments about dissolved Pb was moved to end part of “2.2. Structure and composition of materials”. As for other parts, it is indeed a common writing logic and easy for the reader to read, which was to describe the synthesis and characterization of materials at the beginning. However, two series of materials were synthesized in our paper, and we performed structural characterization with only the best material in each series as a representative because the results of their characterization help us to understand their catalytic behavior. Because of this, we still put the comparison of photocatalytic performance and dark adsorption ability before the structural analysis so that the reader can understand why we did structural analysis for 13.3% BBT-PMO(M) and 16.7% BBT-PMO(N) instead of others.
- Create an overview schematic for figure 1 to give the reader an easily accessible, visual impression of the processes and materials.
Response: We agree with the reviewer. Indeed, we have added Scheme 1 at the end of the introduction part to a visual impression of the processes and materials. See a change in the text (page 2 and 3):
To validate the above strategy, the nanoscale, and the micron-scale lead molybdate (PMO(N) and PMO(M)) were prepared as shown in Scheme 1, and both were wrapped with the BBT via the in-situ synthetic cladding method.
Scheme 1. The synthesis of the BBT-PMO heterojunction and the photocatalytic Cr (VI) reduction over the heterojunction
- Mention the oxidation reaction that accompanies the reduction of Cr6+ to Cr3+.
Response: We agree with the reviewer. Indeed, we consider the oxidation reaction together with the photoreduction of Cr(VI) shown in Figure 11. We emphasize the oxidation reaction in the revised paper. See a change in the text (page 12)
“Therefore, the recombination of the electrons on the LUMO with the hole on the HOMO was effectively suppressed, and then the electrons had more chance to react with Cr(VI) or oxygen. Besides, the hole might be consumed with H2O to produce O2. That is, the photocatalytic reduction activity of the BBT-PMO system under visible light was effectively enhanced for its heterojunction structure.”
- The XPS section should be removed or heavily modified. Probing depth in XPS is usually around 4 nm. In the presented material, the inorganic component seems to be mostly covered by a thick layer of BBT, as seen in Fig. 1 d). Therefore, the probing depth is too shallow to merit any real conclusions, contrary to the definitive statements on the bottom of page 8. If the layer is not too thick, then the metal ions should be visible by XPS and the data should be included in the manuscript. At the moment, the presented conclusion is speculative and not supported by data, which is why it should be removed.
Response: Thank the reviewer’s suggestion.
We have given the XPS spectra of Mo 3d (Figure 8a) and Pb 3d (Figure 8b) in PMO(M), 13.3% BBT-PMO(M), PMO(N) and 16.7% BBT-PMO(N). see page 10:
“To probe the reason for the enhanced photoactivity, the surface elemental composition of BBT-PMO and corresponding PMO substrates were further investigated via the XPS spectra. The high-resolution XPS spectra of Mo 3d, Pb 4f and O 1s were displayed in Figure 8. The binding energy of Mo 3d and Pb 4f in PMO did not change after wrapped by BBT. However, compared to the spectrum of the pristine PMO, a new peak was detected for 13.3% BBT-PMO(M) and 16.7% BBT-PMO(N) at 533.12 eV or at 533.56 eV, which was attributed to the O 1s in the O-C bond [25].”
Figure 8. XPS spectra of (a) Mo 3d, (b) Pb 4f and (c) O 1s in PMO(M), 13.3% BBT-PMO(M), PMO(N) and 16.7% BBT-PMO(N).
- The photocurrent experiment should be described in the methods section.
Response: Thank the reviewer’s comment. We have described the photocurrent experiment in the methods section. See a change in the text (page 14):
“X-ray diffraction (XRD) patterns of the samples were recorded on a Bruker D8 Advance X-ray diffractometer with Cu Kα radiation (λ = 1.5418 nm) and irradiated at a scan rate of 5°/min in the range of 5°-80°. The X-ray tube voltage and current were set at 45 kV and 50 mA, respectively. The morphology and microstructure of the samples were characterized by scanning electron microscopy (SEM JEOL 6700-F) and transmission electron microscopy (TEM, JEOL JEM-2010). The UV-vis diffuse reflectance spectra (DRS) of the prepared samples in the range of 200–800 nm was recorded in the absorption mode using a PerkinElmer lambda 650s UV-vis spectrophotometer. The photocurrent measurement was performed on an electrochemical system (CHI-660D, Chenhua Ltd., China) in a conventional three-electrode configuration with a Pt foil as the counter electrode and a Ag/AgCl (saturated KCl) as the reference electrode. The working electrodes were prepared by the coating method. 10 mg of the as-prepared photocatalyst and 0.5 ml nafion dispersing in 5 mL absolute ethanol (99.5%) were mixed and sonicated for 30 min. The mixture was then spread on a 2.5 cm × 1.0 cm indium-tin oxide glass substrate and dried in air. A 300 W Xe lamp was used as a light source ((λ ≥ 420 nm), and a 0.1 mol/L Na2SO4 aqueous solution was used as the electrolyte. The surface elemental composition of the sample was analyzed by X-ray photoelectron spectroscopy (XPS), which was using an XSAM800 system with Mg Kα X-ray as the excitation source, and all the binding energies were referenced to the C 1s peak at 284.6 eV of the surface amorphous carbon. The Raman spectrum was collected using a confocal microscope, and the wavelength of the excitation laser was 514 nm. The adsorption and desorption isotherms, pore size distribution, and specific surface area of nitrogen were measured at –196 °C using a surface area and pore size analyzer. The generation of ×O2– was detected by Electron Paramagnetic Resonance (EPR, Magnettech MS-5000) with 5, 5-dimethyl-1-pyrroline-N-oxide (DMPO, 100 mM in methanol) as the radical spin-trapped reagent under light irradiation.”
- The photocurrent experiments should be prioritized when discussing the mechanism.
Response: Thank the reviewer’s comment, we have adjusted the writing order, as first described the photocurrent experiments, then XPS experiments, and then UV-vis DRS experiments, and last the Photoreduction rate and ESR experiments. See a change in the text (page 9 and 10)
“The recombination processes of electron-hole pairs in the photocatalytic system have important impacts on its photocatalytic ability. The photocurrent analysis was applied to investigate the photogenerated charge separation capacity and the electron transfer dynamics among BBT, PMO, 13.3% BBT-PMO(M), and 16.7% BBT-PMO(N). As shown in Figure 7, under visible light irradiation, the five samples exhibited a quickly enhanced photocurrent response and remain constant at a comparable high value. In contrast, the photocurrent response intensity remained at a very low value in the dark condition. Moreover, the BBT-PMO composite exhibited higher photocurrent response density compared with the pure BBT and pristine PMO. And the 16.7% BBT-PMO(N) presented the highest photocurrent, which was 2.4 times higher than the 13.3% BBT-PMO(M). The giant photocurrent suggested the 16.7% BBT-PMO(N) obtained greater photogenerated charge separation efficiency, which could lead to higher photocatalytic activity as well. The enhanced photocurrent in the 16.7% BBT-PMO(N) and the 13.3% BBT-PMO(M) compared to the BBT and the PMO, indicating that the heterojunction forms decreased the recombination of photogenerated electron-hole pairs, and therefore more electrons could participate in the photoreduction reaction.”
Figure 7. Photocurrent measurement of (a) BBT, PMO(N) and 16.7% BBT-PMO(N), and (b)BBT, PMO(M) and 13.3% BBT-PMO(M).
“To probe the reason for the enhanced photoactivity, the surface elemental composition of BBT-PMO and corresponding PMO substrates were further investigated via the XPS spectra. The high-resolution XPS spectra of Mo 3d, Pb 4f and O 1s were displayed in Figure 8. The binding energy of Mo 3d and Pb 4f in PMO did not change after wrapped by BBT. However, compared to the spectrum of the pristine PMO, a new peak was detected for 13.3% BBT-PMO(M) and 16.7% BBT-PMO(N) at 533.12 eV or at 533.56 eV, which was attributed to the O 1s in the O-C bond [25]. This indicated that the carbon atoms in the BBT might interact with the oxygen atoms in the PMO at the interface of the two components. This new O-C bond formed after BBT loading on PMO was a sign of the BBT-PMO heterojunction formation. The electronic bonding energy of the O-Pb bond changed from 532.4 eV to 532.0 eV after 16.7% BBT was loaded onto the PMO(N), while it was barely changed for 13.3% BBT-PMO(M) compared to PMO(M) [27]. Likewise, the peak of the O-Mo bond was also changed from 530.05 eV to 530.2 eV after 16.7% BBT was loaded onto the PMO(N), while it was not changed for 13.3% BBT-PMO(M) compared to PMO(M) [27]. These experimental results showed that the BBT component had a more significant effect on the chemical valence state of the surface elements of the PMO(N) than that of the PMO(M) after the formation of the heterojunction, i.e., the BBT and the PMO(N) could form a tighter heterojunction, which might be related to the smaller size and the larger specific surface area of the PMO(N) than the PMO(M).”
|
|
Figure 8. XPS spectra of (a) Mo 3d, (b) Pb 4f and (c) O 1s in PMO(M), 13.3% BBT-PMO(M), PMO(N) and 16.7% BBT-PMO(N).
Equation 3 and 4 need to be referenced to known literature. Also, electronegativity is defined for elements/ions, not compounds. Please specify what X is under these circumstances.
Response: we agree with the reviewer. And we have applied the reference 33 for equations 3 and 4, respectively. Besides, X was defined as the Mulliken electronegativity of PMO, and we also add a sentence for this. See a change in the text (page 11):
“where the X is the Mulliken electronegativity of PMO with the value of 5.98 eV, and EVB and ECB are both vs. NHE [33].”
Please insert the Kubelka Munk formula that was used to interpret Fig. 9.
Response: Following the reviewer’s suggestion, we have inserted the Kubelka Munk formula ((αhν)1/2 = A(hν - Eg)) in Figure 9, See a change in the text (page 11)
Figure 9. (a) UV-vis DRS of BBT, PMO(M), 13.3% BBT-PMO(M), PMO(N), and 16.7% BBT-PMO(N). (b) Tauc plots of BBT, PMO(M), and PMO(N).
Mention and include a reference for the ESR data of O2-.
Response: Following the reviewer’s suggestion, we have inserted reference 25 for ESR data of O2-. See a change in the text (page 12):
“The corresponding EPR analysis (Figure 10B) also confirmed the generation of ×O2- in the photocatalytic systems when the 16.7% BBT-PMO(N) and the 13.3% BBT-PMO(M) were excited by visible light [25].”
Please include a comparison of the leached Pb levels to current thresholds in water as defined by the occupational safety and health administration (USA) or REACH (EU).
Response: we agree with the reviewer that the leached Pd2+ is an intriguing question. Actually, the limit values of lead in drinking water and surface water intended for drinking, as set by EU, USEPA (US Environmental Protection Agency (EPA)) and WHO, are 0.01 mg/L, 0.05 mg/L, and 0.01 mg/L for humans, respectively. While for aquatic life this limited value was 0.1 mg/L, as we listed in our article.
Minor changes:
See attached manuscript.
Response: We apologize for those flaws in the previous manuscript. Following the reviewer’s suggestion, we have revised those minor changes in the text, see the highlight parts.

Round 2
Reviewer 2 Report
The authors addressed my concerns and the manuscript is ready to publish.